# In Vitro Anti-Inflammatory and Immunomodulatory Activities of an Extract from the Roots of *Bupleurum rotundifolium*

**DOI:** 10.3390/medicines6040101

**Published:** 2019-10-11

**Authors:** Juliette Cholet, Caroline Decombat, Marjolaine Vareille-Delarbre, Maël Gainche, Alexandre Berry, François Senejoux, Isabelle Ripoche, Laetitia Delort, Marion Vermerie, Didier Fraisse, Catherine Felgines, Edwige Ranouille, Jean-Yves Berthon, Julien Priam, Etienne Saunier, Albert Tourrette, Yves Troin, Gilles Thebaud, Pierre Chalard, Florence Caldefie-Chezet

**Affiliations:** 1Université Clermont Auvergne, INRA, UNH, Unité de Nutrition Humaine, CRNH Auvergne, F-63000 Clermont-Ferrand, France; caroline.decombat@uca.fr (C.D.); marjolaine.vareille-delarbre@uca.fr (M.V.-D.); alexandre.berry@uca.fr (A.B.); francois.senejoux@uca.fr (F.S.); laetitia.delort@uca.fr (L.D.); marion.vermerie@uca.fr (M.V.); didier.fraisse@uca.fr (D.F.); catherine.felgines@uca.fr (C.F.); gilles.thebaud@uca.fr (G.T.); Florence.CALDEFIE-CHEZET@uca.fr (F.C.-C.); 2SIGMA Clermont, Institut de Chimie de Clermont-Ferrand, BP10448, F-63000 Clermont-Ferrand, France; mael.gainche@sigma-clermont.fr (M.G.); isabelle.ripoche@sigma-clermont.fr (I.R.); yves.troin@sigma-clermont.fr (Y.T.); pierre.chalard@sigma-clermont.fr (P.C.); 3Greentech, Biopôle Clermont-Limagne, 63360 Saint-Beauzire, France; developpement@greentech.fr (E.R.); jeanyvesberthon@greentech.fr (J.-Y.B.); 4Dômes Pharma, 3 rue André Citroën, 63430 Pont-du-Château, France; j.priam@domespharma.com (J.P.); e.saunier@domespharma.com (E.S.); 5AltoPhyto, 7 rue des gargailles, 63370 Lempdes, France; albert.a.tourrette@gmail.com

**Keywords:** *Bupleurum rotundifolium*, PBMCs, cytokines, inflammation, antioxidant

## Abstract

Background: Some Bupleurum species, such as the *Bupleurum chinense* DC. or the *Bupleurum scorzonerifolium* Willd have been extensively studied (especially their roots) for the treatment of inflammation. In contrast, only compounds extracted from the aerial parts of *Bupleurum rotundifolium* have been studied and showed anti-inflammatory or antiproliferative activities. This study was conducted to investigate the antioxidant, anti-inflammatory, and immunomodulatory effects of *Bupleurum rotundifolium* roots. Methods: To tackle the various aspects of inflammation, we studied in vitro a methanolic extract from the roots of *Bupleurum rotundifolium* on peripheral blood mononuclear cells (PBMCs), polymorphonuclear neutrophils (PMNs), and the monocytic cells THP-1. Its antioxidant capacities and iron-chelating activity were assessed. The extract was tested on THP-1 differentiation, reactive oxygen species (ROS) production by leukocytes, neutrophils chemotaxis, cytokines, PGE2 production, and NF-κB activation in PBMCs. Results: The extract showed a decreased ROS production in stimulated cells. It increased PBMC chemokine secretion and up-regulated the differentiation of THP-1 monocytes into macrophage-like cells, indicating a potential interest of the extract in the resolution of acute inflammation. In addition, the analysis of cytokine production suggests that *Bupleurum rotundifolium* has immunomodulatory properties. Conclusions: Cytokines secretion, especially IL-1β and IL-12p70, provided us with a set of indicators suggesting that the extract might be able to drive the polarization of macrophages and lymphocytes toward a Th2 anti-inflammatory profile in excessive inflammation.

## 1. Introduction

Some *Bupleurum* species have been extensively studied as some enter folk medicine. In traditional Asian medicine, “Chaihu”, Bupleuri radix, which refers to the roots of the *Bupleurum chinense* DC. or the *Bupleurum scorzonerifolium* Willd, have been used since at least 200 AD for the treatment of inflammation, fever, and pain. Several studies have confirmed their anti-inflammatory as well as immunomodulatory effects in vitro and in vivo [1]. The main active constituents of these plants, identified as triterpenoid saponins, or saikosaponines, have been extensively studied as well [2]. In contrast, the composition and the properties of *Bupleurum rotundifolium*, which can be found in Europe, have been investigated far less. To our knowledge, only compounds extracted from the aerial parts of *B. rotundifolium* have been studied. These showed anti-inflammatory [3] or antiproliferative activities [4,5], depending on the model used.

Furthermore, the effect of *Bupleurum* species on inflammatory mediators was dependent on the compounds extracted (mainly the type of saikosaponins), the cell type, and the inflammatory mediators measured [6].

Indeed, the complexity of the inflammatory response to injury or infection involves a wide range of cell types (among which are lymphocytes, monocytes, neutrophils, and macrophages) as well as the release of various mediators (prostaglandins, cytokines, kinins, etc.). Polymorphonuclear neutrophils (PMNs) are representative of acute phase of inflammation and account for 50% to 70% of total circulating leukocytes. These immune cells are described as the first cells migrating toward the inflammatory site, thanks to chemotactic gradient, and destroy pathogens, including by releasing reactive oxygen species (ROS). Monocytes and macrophages also act from the beginning, producing ROS, cytokines, chemokines, and prostaglandins though the arachidonic acid pathway. Later, the recruitment of T lymphocytes leads to the extra release of cytokines. The modulation of these mediators will affect the overall results of the inflammation, favoring either the increase of inflammatory response or its resolution. Both may be of interest regarding the context and duration of the process [7,8].

Therefore, the current study was conducted to investigate the potential antioxidant, anti-inflammatory, and immunomodulatory effects of a methanol extract of *Bupleurum rotundifolium* roots. To tackle the various aspects of inflammation, we studied the effects of the extract on peripheral blood mononuclear cells (PBMCs), PMNs, and the human leukemia monocytic cells, THP-1 by means the different markers of inflammation (ROS, monocyte differentiation, chemotaxis, cytokines, COX-2, and PGE2).

## 2. Materials and Methods

### 2.1. Chemicals

All Chemicals were purchased from Sigma-Aldrich (Saint-Quentin-en-Yvelines, France), except for Megabase agarose^®^ (Biorad, Marnes-la-Coquette, France) and dihydrorhodamine 123 (Cayman Chemical Company, Ann Arbor, MI, USA).

### 2.2. Plant Material and Preparation of the Extract

Roots of *Bupleurum rotundifolium* were collected at Chadrat (France) in June 2016. After identification, a voucher specimen was deposited at the University of Clermont Auvergne herbarium (identification number: CLF 110840). After being air-dried, the roots were fine-powdered and macerated in methanol for 24 h three times. After each filtration, the solvent was removed using a rotary evaporator under reduced pressure. The percentage yield of the *Bupleurum rotundifolium* extract (BRE) obtained was found to be 5%.

### 2.3. Major Compounds, Phenolic Content, and Antioxidant Activity

The major constituents of the BRE were determined by comparison with analytical standard (Extrasynthèse, France) or literature data, using liquid chromatography-mass spectrometry (LC-MS, HPLC Ultimate 3000 RSLC chain) and an Orbitrap Q-Exactive (Thermo Scientific, Illkirch, France) with an Uptisphere C18-3 (250 × 4.6 mm, 5 µm) column from Interchim.

The total phenolic content was estimated using the Folin–Ciocalteu method, as previously described by Dubost et al. [9]. A standard curve of gallic acid in the range of 0–300 mg/L was then plotted (R^2^ = 0.9987, y = 0.0033x + 0.244) and the amount of total phenolic compounds was expressed as milligram of gallic acid equivalent (mg GAE) per gram of dry extract. The total flavonoid content was quantified with the Dowd method [10]. Briefly, 1 mL of BRE (0.9955 mg/mL in methanol) was mixed with 5 mL of methanol and 5 mL of a 2% methanolic solution of AlCl3. After 10 min of incubation, the absorbance was measured at 430 nm with a JASCO V-630 spectrophotometer (Jasco, Pfungstadt, Germany). A standard curve of rutin in the range of 1–150 mg/mL was plotted (R^2^ = 0.9987, y = 0.0033x + 0.244), and the amount of total flavonoids was expressed accordingly as milligram of rutin equivalent (mg RE) per gram of dry extract.

The 2,2-diphenyl-1-picrylhydrazyl (DPPH) scavenging activity was evaluated according to the method described by Meda et al. [11], with slight modifications. Briefly, 10 µL of a solution of BRE (1 mg/mL in methanol) was mixed with 2.5 mL of fresh DPPH solution (25 µg/mL in methanol). After 30 min of incubation in the dark, a decrease in DPPH absorbance was recorded at 515 nm using a spectrophotometer (Jasco V-630). A standard curve of Trolox, in the range of 100 µM to 3000 µM, was plotted (R^2^ = 0.9978, y = 0.022x + 1.5186). The results were then expressed as µmol of Trolox equivalents (µmol TE) per gram of dry extract.

An oxygen radical absorbance capacity (ORAC) assay was done in 96-well plates, with a final volume of 200 µL, as described by Gillespie et al. [12]. Fluorescein was used as the fluorescent probe, and 2,2′-azobis (2-methylpropionamidine) dihydrochloride (AAPH) as the peroxyl radical generator. The decrease in fluorescence was measured every minute for 1 h (excitation: 485 nm/emission: 530 nm) using a microplate reader (TECAN infinite F200 PRO, Männedorf, Switzerland). The ORAC value of the BRE was calculated using the regression equation between Trolox equivalents and the net area under the curve (concentration of Trolox in the range of 3–100 μM, R^2^ = 0.9904, y = 35.63x + 11.26). The results were expressed as μmol TE per gram of dry extract.

The iron (II)-chelating activity of the BRE was measured according to Wang et al. [13]. A volume of 100 µL of BRE (5 mg/mL) was blended with 135 µL of distilled water and 5 µL of FeCl_2_ (2 mM) in microplate wells. After 10 min of incubation at room temperature, the reaction was initiated by the addition of 10 µL of ferrozine (5 mM). The decrease of the purple ferrozine–Fe^2+^ complex was estimated 10 min later by monitoring absorbance at 562 nm using the TECAN infinite M200 PRO apparatus (TECAN, Männedorf, Switzerland). Distilled water (100 µL), instead of a sample solution, was used as a control, and 10 µL of distilled water was used for the blank instead of ferrozine solution. A standard curve of ethylenediaminetetraacetic acid (EDTA) in the range of 3 to 50 µg/mL was performed (R^2^ = 0.9832, y = 0.00216x + 0.2676). The results were expressed as µg of EDTA equivalents (µg EDTAE) per gram of dry extract.

### 2.4. Cell Culture and Differentiation

The human monocytic leukemia cell line, THP-1 (American Type Culture Collection, TIB-202^TM^) was cultured and propagated at 37 °C in a humidified atmosphere of 5% CO_2_ in an RPMI 1640 medium (GIBCO, ThermoFisher Scientific, Waltham, MA, USA), which was supplemented with 10% fetal bovine serum (FBS), 2 mM glutamine (Gln), and 50 µg/mL gentamicin.

For differentiation, the THP-1 cells (2 × 10^5^/mL) were incubated in electrosensing plates in a complete growth medium containing 16.2 nM phorbol 12-myristate 13-acetate (PMA) and BRE (0 or 50 µg/mL). The adhesion and the spreading of the cells were monitored continually every 15 min using the real-time cell electronic sensing (RT-CES, ACEA Biosciences Inc., San Diego, CA, USA) system for a period of three days. The RT-CES system detected impedance change derived from cell reactions on the electrodes as the change of cell index (CI).

### 2.5. Leukocyte Viability

Blood was collected from healthy human volunteers (*n* = 6; Etablissement Français du Sang, EFS, Clermont-Ferrand, France). Whole blood leukocytes were obtained by hemolytic shock using ammonium chloride solution (NH_4_Cl, 155 μM; NaHCO_3_ 12 μM, EDTA 0.01 μM). Leukocytes were then washed with Roswell Park Memorial Institute (RPMI), centrifuged (400× *g*, 10 min), and suspended in RPMI. The cell preparations were adjusted to 10^6^ cells/mL with supplemented RPMI (Fetal bovine serum (FBS) 10%, gentamicin 50 μg/mL and Gln 2 mM). The cells were then placed in 96-well polystyrene plates (Cell Wells™, Corning, NY, USA), incubated with BRE at 0 or 50 µg/mL, PMA (0 or 1 µM), and resazurin (25 µg/mL). Fluorescence (excitation/emission: 544/590 nm) was recorded every 30 min for 2 h using the Fluoroskan Ascent FL^®^ apparatus (ThermoFisher Scientific, Illkirch, France).

### 2.6. Kinetics of ROS Production by Leukocytes

The leukocyte preparations (*n* = 6) were obtained as previously described. The cells were placed in 96-well polystyrene plates (Cell Wells™, Corning, NY, USA), incubated with BRE at 0 or 50 µg/mL and dihydrorhodamine 123 (Dh 123, 1 μM), and stimulated or not by 1 µM PMA for 120 min. The fluorescence intensity of rhodamine 123, which is the product of dihydrorhodamine 123 oxidation by ROS, was recorded every 5 min for 120 min (excitation/emission: 485/538 nm) using the Fluoroskan Ascent FL^®^ apparatus.

### 2.7. PBMC Preparation from Human Blood

Blood buffy coats were collected from healthy human volunteers (*n* = 3 volunteers) and carefully layered on a double gradient of Ficoll–Histopaque 1119^®^ and 1077^®^. After centrifugation (400× *g*, 40 min at 20 °C), the first layer of plasma was aspirated, yielding a phase of monocytes and lymphocytes (PBMCs) just above the 1.077 g/mL layer. Then, the Ficoll layer was aspirated, yielding a phase of PMNs corresponding to 1.119 > density > 1.077 g/mL. The phase with PBMCs was washed with RPMI and centrifuged (5 min, 400× *g*) twice, and then suspended in 5 mL of supplemented RPMI (FBS 10%, gentamicin 50 μg/mL, and Gln 2 mM). Meanwhile, the residual erythrocytes in the PMNs phase were lyzed by hemolytic shock using ammonium chloride solution. The PMNs were then washed with RPMI, centrifuged, and suspended in 1 mL of supplemented RPMI. Both cell preparations were adjusted to 10^6^ cells/mL for assays.

### 2.8. Chemotaxis of PMNs Incubated with BRE

Chemotaxis was determined by agarose assay (*n* = 4 volunteers) as previously described [14]. Eight sets of three 2.5 mm diameter wells placed 2.5 mm apart were cut into agarose gel (Megabase agarose^®^). To determine the effect of BRE on PMN chemotaxis towards a chemoattractant agent (formyl-methionyl-leucyl phenylalanine, fMLP), the PMNs (10^6^ cells/mL) were incubated with BRE at 0 or 50 µg/mL. After centrifugation, the concentrated PMNs (7.5 × 10^5^ cells/5 μL) were placed in the central wells alongside the chemotactic factor fMLP (0.1 µM) in the outer wells and the culture medium in the inner wells. In this assay, the plates were incubated for 90 min (37 °C, 5% CO_2_). Migration was measured with an ocular micrometer (magnification ×40) as the distance between the border of the well in the middle and the leading edge in the direction of the chemoattractant well (directed migration, DM, μm) or of the control well (spontaneous migration, SM, µm). The results were expressed as the ratio between mean DM and mean SM of four assays.

### 2.9. Enzyme-Linked Immunosorbent Assay (ELISA)

PBMCs (10^6^ cells/mL) (*n* = 2 volunteers) were incubated with or without lipopolysaccharide (LPS) (1 µg/mL, LPS O26:B26, Sigma-Aldrich) and BRE (0 or 50 µg/mL) for 24 h. The PGE2 and the COX-2 in the culture media were assessed by ELISA, using the PGE2 assay kit from Invitrogen (Invitrogen™; Thermo Fisher Scientific; Waltham, MA, USA) and the COX-2 assay kit from Elabscience (Houston, TX, USA).

### 2.10. Determination of Cytokine Concentrations

PBMCs (10^6^ cells/mL) (*n* = 3 volunteers) were incubated with or without phytohemagglutinin (PHA, 5 µg/mL) and BRE (0 or 50 µg/mL) for 24 h. Milliplex^®^ MAP kits (EMD Millipore™; Merck; Darmstadt, Germany) were used for all assays. All samples were run in triplicate and were assayed for 14 human cytokines (IFNγ, IL-10, IL-12 p70, IL-1β, IL-2, IL-21, IL-4, IL-23, IL-5, IL-6, IL-8, MIP-1α, MIP-1β, and TNFα). Cytokine levels were measured using optimal concentrations of standards and antibodies according to the manufacturer’s instructions. After completion of all the steps in the assay, the plates were read in the Luminex Bio-Plex 200 System (Biorad, Marnes-la-Coquette, France) and the data analyzed using BioPlex Manager™ 4.1 software with a five-parameter logistic regression (5PL) curve fitting.

### 2.11. Western Blot Analysis for NF-κB

Determination of NF-κB protein was performed with the protein extract from PBMCs (*n* = 1 volunteer) by Western blotting. The PBMCs were incubated with or without LPS (1 µg/mL) and BRE (0 or 50 µg/mL) for 1 h. Thirty micrograms of the proteins were separated by electrophoresis in a 10% polyacrylamide gel and transferred at 4 °C to a polyvinylidene membrane (Biorad). Immunoblots were blocked with TBS–Tween-20 0.1% containing 5% dry milk, and then probed with a primary antibody (NF-κB p65, Cell Signaling Technology, Leiden, The Netherlands). The immunoblots were incubated with a horseradish peroxidase-conjugated secondary antibody (Cell Signaling Technology). The immune reactive strips were visualized by chemiluminescence (Pierce ECL Western Blotting Substrate, Thermo Fisher Scientific). Band densities were quantified using Fiji [15]. β-actin (Cell Signaling Technology) was used as an internal control on each gel to normalize signal intensities between gels.

### 2.12. Statistical Analysis

Data were expressed as mean ± SD. Statistical analysis was performed using R software (version 3.4.2). The normality of the variables was assessed by the Shapiro–Wilk test, and their homoscedasticity by Bartlett’s test. Multiple comparisons between groups were made by the Kruskal–Wallis test followed by Dunn’s test. Comparisons between the two groups were made by the Student test or the Mann–Whitney test when normality was rejected. Values with *p* < 0.05 were considered significant.

## 3. Results

### 3.1. BRE Content and Antioxidant Properties

The analysis of the main compounds of the extract showed the presence of organic acids (quinic acid and citric acid), saccharose, esters of quinic acid and cinnamic acid (chlorogenic acid, cryptochlorogenic acid, and feruloylquinic acids), and flavonoid glycosides (isoquercitrin, narcissin and isorhamnetin-3-*O*-glucoside) (Table 1). The radical scavenging capacity of the extract was determined using DPPH and ORAC assays. The results, along with the iron (II)-chelating activity and total phenolic and flavonoid contents, are shown in Table 2. ORAC and DPPH values, reflecting the direct antioxidant activity of the BRE, were found to be weak (1277 ± 852 and 198 ± 8.5 µM TE). Its capacity to scavenge free ferrous ion was low as well, with 47.3 ± 3.2 µg EDTAE. Its total phenolic content was estimated at 47 ± 0.3 mg GAE, of which 30% flavonoids (13.9 ± 0.2 mg GAE).

### 3.2. BRE Inhibited ROS Production of Blood Leukocytes

To investigate further the potential anti-inflammatory and antioxidant effects of BRE, we examined its effect on blood leukocytes ROS production triggered by PMA. PMA stimulation resulted in a significant increase in ROS production after 1 h of incubation (Figure 1a,b). Incubation with BRE inhibited the production in a dose-dependent manner. The decrease was significant after 1 h for a 100 µg/mL dose (>30%). After 2 h, significance was obtained for a 25 µg/mL dose and above (−26.5%, −34.3%, and −48.8% for 25, 50, and 100 µg/mL, respectively) (Figure 1c). This effect was not the consequence of a decrease in cell viability or proliferation, as the viability assay did not show any significant difference between cells incubated without or with BRE in the concentration range 10–100 µg/mL after 2 h (Figure 2). The BRE had no effect on ROS production in nonstimulated cells (data not shown).

### 3.3. BRE Enhanced PMNs Chemotactic Index

To confirm the anti-inflammatory effect of BRE, we analyzed its impact on PMN migration toward a chemoattractant. In the presence of BRE (50 µg/mL), PMN migration toward the culture medium (SM) was decreased by more than half (31.3 ± 9 vs. 75 ± 22, *p* < 0.05), whereas no significant change was observed for DM (migration toward chemoattractant). Thus, the chemotactic index (DM/SM) was higher with the BRE (Table 3).

### 3.4. BRE Promoted the Differentiation of THP-1 Cells

As shown in Figure 3, incubation with PMA increased the adhesion and spreading of THP-1 cells in the wells, reflecting their differentiation from monocyte to macrophage. Without BRE, impedance reached a plateau after around 60 h of incubation. In the presence of BRE (50 µg/mL), the slope of the curve was higher, and impedance started to be significantly different from control after around 40 h of incubation (*p* < 0.05). This suggests that, overall, the BRE sped up the differentiation of THP-1 to macrophage-like cells.

### 3.5. BRE Increased PGE2 Concentration without LPS

Incubation with BRE at 50 µg/mL did not change the concentration of COX-2 (data not shown) nor PGE2 in the supernatants of PBMCs stimulated by LPS for 24 h (Figure 4). However, we observed that in cells incubated with BRE alone, PGE2 concentration was 384% higher (198.8 ± 6 vs. 41 ± 13 pg/mL) than in the supernatants of the control cells (without LPS and without BRE, *p* < 0.05).

### 3.6. BRE Impacted PBMCs Cytokines Secretion

Despite a downward trend, BRE at 50 µg/mL was not able to reach statistical significance concerning IL-2 secretion due to high variations in nonstimulated or stimulated PBMCs (4.5 ± 0.7 vs. 15.4 ± 11 pg/mL and 125.6 ± 101 vs. 327.9 ± 188 pg/mL, respectively) (Table 4).

The BRE heightened the secretion of IL-4 in nonstimulated cells (×2.5, *p* = 0.03), but it was not affected in stimulated cells. Also, the BRE did not impact the secretion of IL-10 and IL-5. What is more, IL-5 could not be detected in the supernatants of nonstimulated cells (Table 4).

Looking at pro-inflammatory cytokines, we observed that the BRE diminished the production of IL-12p70 by more than half in PHA-stimulated PBMCs (17.1 ± 10.9 vs. 40.7 ± 11.9 pg/mL, *p* = 0.05) (Table 4). Moreover, the ratio between IL-12p70 and IL-10 remained stable in nonstimulated PBMCs but tended to be decreased by the BRE in the presence of PHA (0.001 ± 0.0009 vs. 0.003 ± 0.003, *p* = 0.08), indicating a potential anti-inflammatory effect of the extract. The BRE tended to increase the secretion of IL-1β in nonstimulated PBMCs, but the change was not significant due to important inter-individual variations (952.1 ± 890 vs. 9.2 ± 9.1 pg/mL, *p* = 0.09). In PHA-stimulated cells, it reduced by half the concentration of IL-1β (1334.7 ± 33.5 vs. 2548.5 ± 402.2 pg/mL, *p* < 0.01). On the contrary, it strongly increased TNFα secretion, with or without PHA stimulation (×45 and ×1.6, respectively).

The secretion of chemokines IL-8, MIP-1α, and MIP-1β were significantly increased in the presence of BRE, both in PHA-stimulated cells for IL-8 and in nonstimulated ones for MIP-1α and MIP-1β. IL-8 production tended to be increased in nonstimulated cells as well, but failed to reach significance (816.4 ± 5.6 vs. 535.7 ± 196 pg/mL, *p* = 0.07).

Figure 5 is a summary of the global effect of BRE on the main pro- and anti-inflammatory cytokines in PHA-stimulated PBMCs. It can be observed that the BRE had a conservative effect on the anti-inflammatory cytokines profile (right side of the figure), whereas it modified the behavior of the pro-inflammatory ones (left side of the figure).

### 3.7. BRE Showed a Dual Effect on NF-κB Activation

The BRE effect on NF-κB activation was different depending on the presence or the absence of LPS (Figure 6). Whereas NF-κB p65 fragment could not be detected in the control cells (0 µg/mL BRE, without LPS), a slight amount was revealed on cells incubated with BRE but without LPS. Conversely, the presence of BRE diminished the activation of NF-κB in cells stimulated with LPS.

## 4. Discussion

*Bupleurum rotundifolium* extract revealed to have a weak ROS scavenging activity: 1277 µmol TE/g for the ORAC assay and 198 µmol TE per gram of dry extract for the DPPH assay. By comparison, previous studies with similar methods found that the ORAC values of green teas, which are often used as references, can range from 2730 to 5031 μmol TE/g and DPPH values from 1098 to 1376 μmol TE/g [16]. These results are not surprising when considering the low phenolic content, especially flavonoids [17]. The scavenging activity of *Bupleurum* has shown to be very variable, depending on the considered species. Gevrenova et al. obtained contrasted results with DPPH assay by testing *B. flavum*, *B. Baldense*, and *B. affine*, and results were well correlated with the phenolic content of the extracts [18]. As for the traditional Chinese medicinal plants *B. scorzonerifolium* and *B. chinense*, they exhibited a low DPPH value (75.3 µmol TE per 100 g of dry extract and 19.93 µmol TE per g of dry extract, respectively) [19,20].

In addition to the DPPH and ORAC tests, which gave direct antioxidant activity, we measured the iron (II)-chelating activity of BRE. Indeed, free ferrous iron is involved in the Fenton reaction, which is an important generator of ROS in vivo. As seen in previous assays, the BRE iron-chelating capacity was found to be low in comparison with other plant extracts [21]. Nonetheless, we decided to investigate the potential anti-inflammatory activity of BRE in vitro by studying the different mediators of cell inflammatory response. First, it was established that BRE at 100 µg/mL reduced ROS production significantly after 1 h of incubation with stimulated leukocytes, and at 25, 50, and 100 µg/mL after 2 h, without affecting the viability of cells. With regard to the very slight direct antioxidant potential of the extract, it could be suggested that the BRE indirectly reduced ROS production, mainly by affecting the enzyme activities implied in ROS generation (protein kinase C, NADPH oxidase) or ROS detoxification (catalase, superoxide dismutase, etc.), rather than by directly scavenging reactive species. This protective effect against oxidative stress has been observed in other *Bupleurum* species. Despite the fact that the analysis of the major constituents of our extract did not reveal triterpenoid saponins, it cannot be excluded that they may be present at lower concentrations, as some have been detected in the aerial parts of *B. rotundifolium* [3]. In particular, it has been shown that saikosaponin D extracted from the roots of *Bupleurum falcatum* increased the activity of the main antioxidant enzymes (i.e., superoxide dismutase (SOD), catalase (CAT), and glutathione peroxidase) [22]. Similarly, an aqueous extract from the roots of *B. falcatum* increased the activities of SOD and CAT in a L-thyroxine-induced rat model [23]. Moreover, it has been shown that the main compounds of the BRE, especially flavonoid glycosides could be involved, at least in part, in the protective effect against ROS toxicity [24].

Furthermore, it is worthy to note the decrease of IL-1β in stimulated PBMCs when incubated with BRE, whereas there was again no effect on cell proliferation regardless of the cell line used. This can be observed though the stability of IL-2 production of PBMCs and viability of total leukocytes or of THP-1 cells after 24 h incubation. It has been shown that the production of IL-1β was ROS-dependent, as they allowed the activation of inflammasome NLRP3, which, via caspase-1, turns proIL-1β to active IL-1β [25,26,27]. The secretion of IL-1β also depends on the activation of the NF-κB pathway. Its activation triggers the secretion of TNFα as well [28]. Given that in the same conditions, the secretion of TNFα is increased in both nonstimulated and stimulated cells by the extract, we hypothesized that BRE does not prevent NF-κB pathway activation and that it could even stimulate it. This hypothesis was supported by the fact that in nonstimulated PBMCs, whilst ROS production was not affected by BRE incubation, IL-1β secretion tended to increase. Our results on detection of NF-κB activation by Western blot tends to confirm this hypothesis. Indeed, we were able to detect NF-κB activation by the BRE alone. However, the effect of BRE on this pathway does not seems not entirely clear yet, as it showed the opposite effect in LPS-stimulated cells, dampening by half the presence of NF-κB p65 fragment. This last observation is in line with what has been observed with the best-known *Bupleurum* species. For example, Zhu et al. established that saikosaponin A, found in the roots of *B. chinense* or *B. scorzonerifolium* suppressed the LPS-induced activation of NF-κB in murine macrophages [29]. Similarly, Bremner et al. isolated NF-κB inhibitory compounds from aerial parts of *B. fruticosum* [30].

The efficiency of BRE on PMN chemotaxis is questionable, as the increase of the chemotactic index did not reflect an augmented migration toward fMLP. Hence, it cannot be concluded that BRE stimulates processes underlying adhesion and movements of PMNs. In a previous study, similar results were obtained with saikosaponin D extracted from *Bupleurum* radix. It increased random migration of macrophages but did not direct migration [31]. Nonetheless, concerning our extract, the profile of chemokine secretion (i.e., IL-8, MIP-1α, and MIP-1β) by PBMCs suggest that by increasing the release of chemotactic factors, it could enhance leukocyte migration toward the inflammatory site [32,33,34].

In addition, the increase of chemokines in nonstimulated cells supports our results of an activation of the NF-κB pathway by the BRE. Indeed, the increase in MIP-1α also indicates that the BRE acts upstream to NF-κB in the Toll-like receptor signaling pathway [34,35].

Intriguingly, COX-2 induction was not affected by BRE in either nonstimulated or LPS-stimulated cells, whereas PGE2 production increased in nonstimulated cells. One hypothesis is that the duration of incubation does not allow the detection of changes in COX-2 concentrations. Indeed, as COX-2 exhibits a short half-life, an increase at the beginning of the assay cannot be detected after 24 h of stimulation with LPS [36].

Be that as it may, the increase of PGE2 and IL-4 suggests that BRE may potentially orient lymphocytes toward a Th2 profile in a physiological situation [37,38]. Furthermore, our results on THP-1 show that BRE can up-regulate the differentiation of monocytes into macrophages in vitro. These results, along with the increase of chemokines, suggest that BRE might help improve the resolution of acute inflammation. In the case of excessive inflammation, the decrease of the IL-1β, IL-12p70, and IL12p70/IL-10 ratio suggest an anti-inflammatory effect of the BRE, which may drive T cells toward a Th2 profile [37,39,40,41].

## Figures and Tables

**Figure 1 medicines-06-00101-f001:**
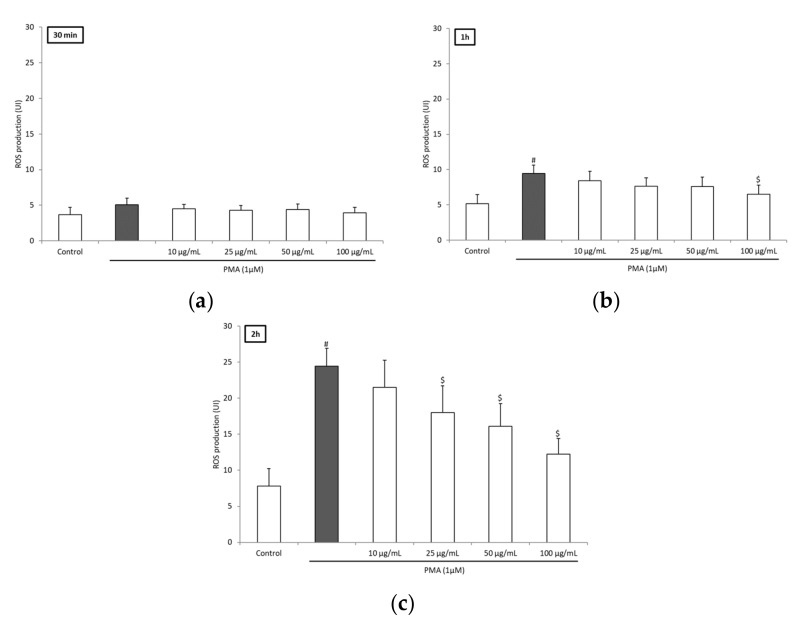
BRE concentration dependently inhibited ROS production of blood leukocytes. Cells were incubated with BRE (0, 10, 25, 50, and 100 µg/mL) and stimulated with PMA (1 µM) for (**a**) 30 min; (**b**) 1 h; and (**c**) 2 h. Data were shown as means ± SD; # *p* < 0.05 compared with Control, $ *p* < 0.05 compared with PMA-stimulated cells (0 µg/mL BRE). BRE: *Bupleurum rotundifolium.*

**Figure 2 medicines-06-00101-f002:**
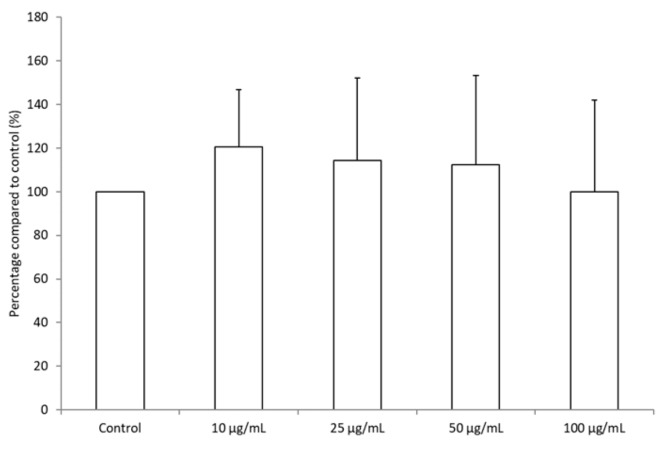
BRE did not affect leukocyte viability. Cells were treated with the indicated concentrations of BRE for 2 h, and then cell viability was measured. Data were shown as means ± SD (Control = 100%); * *p* < 0.05 compared with Control.

**Figure 3 medicines-06-00101-f003:**
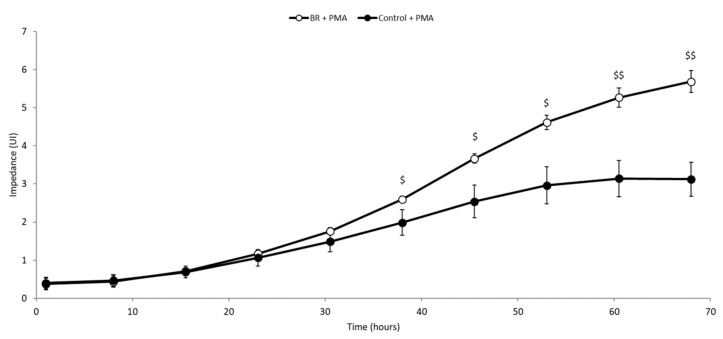
BRE promoted the differentiation of THP-1 cells. Cells were incubated with BRE (50 µg/mL) and stimulated with PMA (16.2 nM) for 70 h. $ *p* < 0.05 compared with PMA-stimulated cells, $$ *p* < 0.01 compared with PMA-stimulated cells.

**Figure 4 medicines-06-00101-f004:**
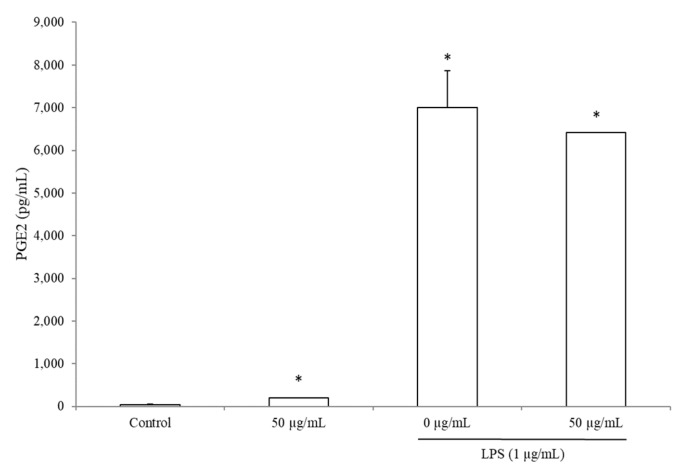
BRE increased PGE2 production without LPS. Cells were incubated with BRE as indicated and stimulated with LPS (1 µg/mL) for 24 h. Data were shown as means ± SD; * *p* < 0.05 compared with Control (paired *t*-test).

**Figure 5 medicines-06-00101-f005:**
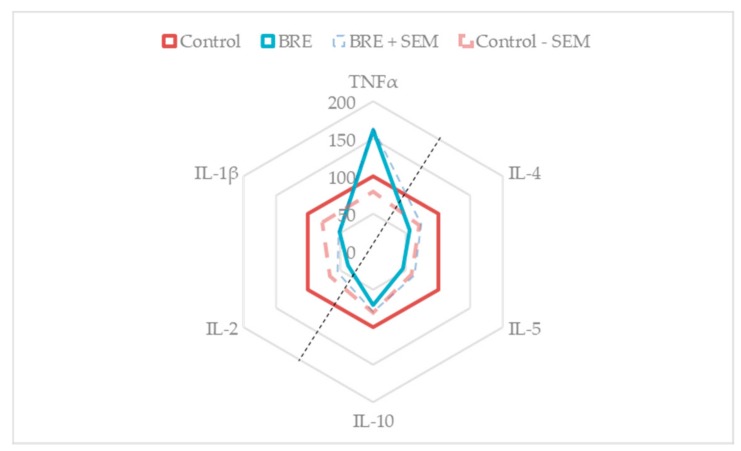
BRE impacted cytokines secretions of PBMCs. Cells were incubated with BRE as indicated and stimulated with PHA (5 µg/mL) for 24 h. Red solid line: production of main pro- and anti-inflammatory cytokines in Control (PHA-stimulated cells, 0 µg/mL BRE). Blue solid line: production of main pro- and anti-inflammatory cytokines in supernatants of cells incubated with BRE and PHA. Data were shown as means (Control = 100%). * *p* < 0.05 compared with Control. Red dotted line: Control mean minus corresponding standard error mean. Blue dotted line: BRE and PHA mean plus corresponding SEM.

**Figure 6 medicines-06-00101-f006:**
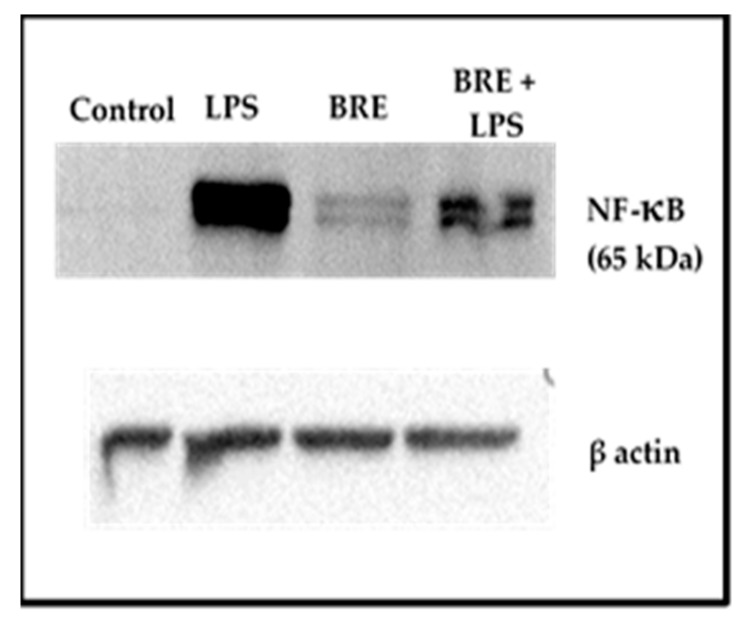
BRE increased NF-κB p65 in PBMCs. Cells were incubated with BRE (0, or 50 µg/mL) and stimulated with LPS (1 µg/mL) for 1 h.

**Table 1 medicines-06-00101-t001:** Some of the major constituents of the methanolic extract of *Bupleurum rotundifolium* roots.

RT ^1^ (min)	[M − H]^−^ (*m*/*z*)	Compound
3.84	191.0549	Quinic acid ^2^
4.1	341.1085	Saccharose ^2^
6.83	191.0186	Citric acid ^2^
12.92	353.0875	Chlorogenic acid ^2^
14.31	353.0875	Cryptochlorogenic acid ^2^
15.88	367.1029	3-*O*-feruloylquinic acid ^3^
17.52	367.1030	Feruloylquinic acid Isomer ^3^
18.46	463.0880	Isoquercitrin ^2^
20.72	623.1614	Narcissin ^3^
22.95	477.1035	Isorhamnetin-3-*O*-glucoside ^3^

^1^ Retention time. ^2^ Identification by comparison with an analytical standard. ^3^ Identification by comparison with publication.

**Table 2 medicines-06-00101-t002:** Radical scavenging capacity, iron (II)-chelating activity, and phenolic and flavonoid contents.^1^.

ORAC (µM TE ^2^)	DPPH (µM TE ^2^)	Iron (II) Chelation (µM EDTAE ^3^)	Flavonoid Content (mg RE ^4^)	Total Phenolic Content (mg GAE ^5^)
1277 ± 852	198 ± 8.5	47.3 ± 3.23	13.9 ± 0.2	47 ± 0.3

^1^ Data are expressed as the mean of triplicate ± SD. ^2^ TE: Trolox equivalent. ^3^ EDTAE: EDTA equivalent. ^4^ RE: Rutin equivalent. ^5^ GAE: Gallic acid equivalent.

**Table 3 medicines-06-00101-t003:** Influence of BRE on PMNs chemotaxis ^1^.

Migration	BRE Concentration (µg/mL)	
	0	50
SM ^2^	75 ± 22	31.3 ± 9 *
DM ^3^	90 ± 24	67.5 ± 7
DM/SM	1.3 ± 0.4	2.3 ± 0.6 *

^1^ Data are expressed as mean ± SD, * *p* < 0.05 compared with control (0 µg/mL BRE). ^2^ SM: spontaneous migration toward culture media ^3^ DM: directed migration toward chemotactic factor (fMLP). PMNs: Polymorphonuclear neutrophils.

**Table 4 medicines-06-00101-t004:** BRE impacted cytokines secretions of PBMCs. Cells were incubated with BRE as indicated and stimulated with PHA (5 µg/mL) for 24 h. Data were shown as means ± SD (pg/mL); ND: not detected.

	Without PHA			PHA (5 µg/mL)		
	BRE Concentration (µg/mL)			BRE Concentration (µg/mL)		
	0	50	Variation ^1^	0	50	Variation ^1^
IL-2	15.4 ± 11	4.5 ± 0.7	NS	327.9 ± 188	125.6 ± 101.3	NS
IL-4	1.4 ± 1	3.5 ± 0.9	(++)	198.7 ± 96.6	112.2 ± 28.8	NS
IL-10	14.4 ± 7.3	10.6 ± 6.1	NS	10,354.5 ± 3830	10,284.7 ± 1350	NS
IL-5	ND	ND		33.9 ± 23.1	15.37 ± 11.9	NS
IL12p70	0.73 ± 0.6	0.24 ± 0.3	NS	40.73 ± 11.9	17.1 ± 10.5	(−−)
IL-1β	9.2 ± 9.1	952 ± 890	(+)	2548.5 ± 402.2	1334.7 ± 33.5	(−−)
TNFα	15.9 ± 9.2	715 ± 216	(++)	4637.2 ± 1518	7538.7 ± 185.5	(++)
IL-8	535.7 ± 196	816 ± 5.8	(+)	748.9 ± 65	1197 ± 19	(++)
MIP1α	4.2 ± 2.7	204 ± 163	(+)	1757.8 ± 578	1026.1 ± 20.7	(−)
MIP1β	137 ± 107	3300 ± 2890	(++)	6400.1 ± 603.5	6275.6 ± 359.2	NS
IL-12p70/ IL-10	0.05 ± 0.03	0.03 ± 0.05	NS	0.001 ± 0.0009	0.003 ± 0.003	(−)

^1^ (+): positive change between 0 and 50 µg/mL BRE (trend, 0.05 < *p* < 0.1); (++): significant positive change between 0 and 50 µg/mL BRE (*p* < 0.05); (−): negative change between 0 and 50 µg/mL BRE (trend, 0.05 < *p* < 0.1); (−−): significant negative change between 0 and 50 µg/mL BRE (*p* < 0.05); NS: no significant difference (*t*-test). PBMCs: peripheral blood mononuclear cells.

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
