# Peer review of "In Vitro Anti-Inflammatory and Immunomodulatory Activities of an Extract from the Roots of Bupleurum rotundifolium"

_medicines, 2019, doi:10.3390/medicines6040101_

Round 1
Reviewer 1 Report
The authors got extracts of Bupleurum rotundifolium root, and measure the total phenolic content and total flavonoids of BRE. The study investifate the antioxidant, anti-inflammatory and immunomodulatory effects of extract, such as ROS, NF-kB, THP-1, and something like that. The manuscript also discussed the probably pharmacological pathway of the extract. So the paper should be accepted after minor revision.
Question:
1.Do you plan to figure out what compounds BRE has? Compare BRE extract’s compounds with the compounds of aerial parts of B. rotundifolium or compounds of Burpleurum chinense DC or Bupleurum scorzonerifolium In previous study should be interesting.
2.The study compared the ROS scavenging activity of BRE and green teas and suggested the extract should be indirectly reduced ROS production by affecting the enzyme activities implied in ROS generation or ROS detoxification, rather than directly scavenging reactive species in discussion. This is a good discussion orientation, but why not compared with other Burpleurum? That’s should be more comparability.
3.In Figure 3, the curve of Control+PMA has already reached a plateau at 60 hrs, but the curve of BRE+PMA still has a significant increase trend at 70 hrs, why not incubate them for longer time to see when they reach the plateau?
4.BRE did not change the concentration of COX-2 nor PGE2 in the supernatants of PBMCs stimulated b LPS for 24 hrs. Why Figure 4 just shows the data of PGE2, but not show the data of COX-2?
5.In figure 6, the bands of BRE+LPS was interrupted in western blot, the data can’t be used, especially for quantification, please replace the image with a perfect one.
Author Response
Dear Madam/dear Sir,
Thank you for your comments about our manuscript.
Regarding extract’s compounds, our team recently identified some of the major compounds of the extract, so we added it in the manuscript.
Secondly, the comparison with other Bupleurum species for scavenging activity is indeed an interesting point that we forgot to add in this discussion. We first choose green tea as it is likely a standard for antioxidant assay. Based on your comment, we added some lines and references in the discussion to allow comparison with other species.
Regarding point number 3, we chose not to extend further to 72h, because of the mortality we observed when culturing THP-1. This cell line can grow and survive in the same culture media, at appropriate cell concentration until 72h, but beyond we start to observe mortality. In addition, we cannot change the media in the Icelligence apparatus without to panic the sensor. It hence appeared to us that it was better not to incubate longer the cells.
Point number 4: we tested supernatants after 24h of incubation; however, as mention in the discussion part, we believe that the assay of COX-2 at this time might not be relevant. The difference between control cells and stimulated cells was surprisingly low. We have recently tried to determine the kinetics of COX-2/PGE2 following stimulation by LPS. The production of PGE2 really started to increase between 6 and 12h and was at his maximum after 24h, but gene expression of COX-2 reached a peak after 4h of stimulation. We were not able to determine the peak of COX-2 concentration due to a flaw in the new assay kit we received (and we are waiting to receive a new one), however it can hypothesized that it has to be done before 24h.
Point number 5: we agree we your remark on the workability of this image for quantification. Sadly, the other images were performed with lower protein amount and thus are not workable either. Thus, we propose not to use this image for quantification, and we removed the graph from the manuscript.
Reviewer 2 Report
This paper is interesting research on popular topic – antioxidant/anti-inflammatory effect of natural extracts.Very well prepared introduction section which shows general aim and idea of this study.
Main limitation of this paper is lack of chemical characterization of tested extract, in my opinion without that the manuscript will not have full scientific value. Second in low number of independent tests (n= 2,3 or 4 in my opinion is too low to obtain clear results on blood donors, the 6 should be minimum)
ATCC number of cell line should be provided
Lines 145 and 156 lack of upper index
Results should be expressed as mean with SD not SEM
Figure 1b has some error in resolution
According to GLP and guidelines for clinical studies when Authors performed test on human subject (even on healthy donors), they should give number of ethic commission agreement on that.
Author Response
Dear Madam/dear Sir,
Thank you for your comments about our manuscript.
Regarding extract’s compounds, our team recently identified some of the major compounds of the extract, so we added it in the manuscript.
As for the low number of tests, we agree and are aware of the limitations that it implies. We tried our best not to over-interpreted the results in line with the high variability we encountered in some of the assays.
With regards to your corrections, we modified the manuscript accordingly. In particular, SD and SEM have different interpretations, and we admit that we “routinely” use the SEM rather than the SD. There is not a big step between the two (as you pointed out, we had a low number of replicate), however it is indeed fairer in this case to use SD and we changed it in the manuscript.
Regarding your comment on GLP and guidelines for clinical studies: after consulting with the EFS manager in charge of ethical questions, they indicated that in our case the French articles cited below were sufficient, as we harvested blood samples from anonymous donors who were not enrolled for the purpose of this study in a trial.
In addition, we can cite the following article from French Public Health Code (n° L1221-8-1 ), which states that "Blood and its components may be used in the context of a research activity, whether or not they have been collected within the Etablissement Français du sang [EFS, French Blood Establishment]. In this case, the research is conducted on samples taken either for medical purposes or as part of research involving the human person.[...]". I hope this answers your concerns about ethical guidelines.
Round 2
Reviewer 2 Report
The major issue related to chemical characterisation has been corrected, other comments also have been included in revised version.